# HOTAIR Promotes the Hyperactivation of PI3K/Akt and Wnt/β-Catenin Signaling Pathways via PTEN Hypermethylation in Cervical Cancer

**DOI:** 10.3390/cells13171484

**Published:** 2024-09-04

**Authors:** Samuel Trujano-Camacho, David Cantú-de León, Eloy Pérez-Yepez, Carlos Contreras-Romero, Jossimar Coronel-Hernandez, Oliver Millan-Catalan, Mauricio Rodríguez-Dorantes, Cesar López-Camarillo, Concepción Gutiérrez-Ruiz, Nadia Jacobo-Herrera, Carlos Pérez-Plasencia

**Affiliations:** 1Experimental Biology PhD Program, DCBS, Universidad Autónoma Metropolitana-Iztapalapa, Mexico City 09340, Mexico; samuel.trujano1@gmail.com; 2Laboratorio de Genómica, Instituto Nacional de Cancerología, Av. San Fernando 22, Belisario Domínguez Secc 16, Tlalpan, Ciudad de México 14080, Mexico; dfcantu@gmail.com (D.C.-d.L.); eperezy2306@gmail.com (E.P.-Y.); carlos.oncogenomica@gmail.com (C.C.-R.); jossithunders@gmail.com (J.C.-H.); oliver.millan.sg@gmail.com (O.M.-C.); 3Laboratorio de Oncogenómica, Instituto Nacional de Medicina Genómica (INMEGEN), Mexico City 14610, Mexico; mrodriguez@inmegen.gob.mx; 4Posgrado en Ciencias Genómicas, Universidad Autónoma de la Ciudad de México, Ciudad de México 03100, Mexico; cesar.lopez@uacm.edu.mx; 5Laboratory of Experimental Medicine, Translational Medicine Unit, Instituto de Investigaciones Biomédicas, UNAM/Instituto Nacional de Cardiología Ignacio Chávez, Tlalpan, Mexico City 14080, Mexico; mcgr@xanum.uam.mx; 6Department of Health Sciences, Universidad Autónoma Metropolitana-Iztapalapa, Mexico City 09340, Mexico; 7Unidad de Bioquímica, Instituto Nacional de Ciencias Médicas y Nutrición Salvador Zubiran, Av. Vasco de Quiroga 15, Col. Belisario Domínguez Sección XVI, Tlalpan, Ciudad de México 14080, Mexico; nadia.jacoboh@incmnsz.mx; 8Unidad de Biomedicina, Facultad de Estudios Superiores Iztacala, Universidad Nacional Autónoma de México (UNAM), Tlalnepantla 54090, Mexico

**Keywords:** LncRNAs, HOTAIR, epigenetic regulation, PI3K/AKT pathway, Wntl/β-catenin pathway

## Abstract

The mechanisms underlying the sustained activation of the PI3K/AKT and Wnt/β-catenin pathways mediated by HOTAIR in cervical cancer (CC) have not been extensively described. To address this knowledge gap in the literature, we explored the interactions between these pathways by driving HOTAIR expression levels in HeLa cells. Our findings reveal that HOTAIR is a key regulator in sustaining the activation of both signaling pathways. Specifically, altering HOTAIR expression—either by knockdown or overexpression—significantly influenced the transcriptional activity of the PI3K/AKT and Wnt/β-catenin pathways. Additionally, we discovered that HIF1α directly induces HOTAIR transcription, which in turn leads to the epigenetic silencing of the PTEN promoter via DNMT1. This process leads to the sustained activation of both pathways, highlighting a novel regulatory axis involving HOTAIR and HIF1α in cervical cancer. Our results suggest a new model in which HOTAIR sustains reciprocal activation of the PI3K/AKT and Wnt/β-catenin pathways through the HOTAIR/HIF1α axis, thereby contributing to the oncogenic phenotype of cervical cancer.

## 1. Introduction

Recently, the role of long non-coding RNAs (LncRNAs) as essential effectors of signaling pathways in cancer development has been extensively explored. LncRNAs are RNA molecules longer than 200 base pairs, primarily transcribed by RNA polymerase II, featuring a 5’ cap and a 3’polyadenylated tail. Their intricate structure enables them to regulate transcriptional and epigenetic mechanisms by interacting with DNA, RNA, and proteins. This regulatory capacity allows LncRNAs to sustain signaling pathways such as PI3K/AKT and Wnt/β-catenin, thereby promoting the progression of cancers such as cervical cancer (CC). These insights underscore the pivotal role of LncRNAs in cancer biology and highlight their potential as targets for therapeutic interventions [1,2,3].

LncRNAs such as LINC01133 and LINK-A have been reported as effectors of the PI3K/AKT pathway regulating mTORC2 by stabilizing its messenger and through direct interaction with PIP3, respectively [4,5]. Similarly, NEAT1 has been described as a regulator of PI3K through the downregulation of miRNAs such as miR-1294 and miR-17 in gastric cancer [6,7].

Concerning the Wnt/β-catenin pathway, LncRNAs such as LINC01606 and NEAT1 exert their function through negative regulation of miR-423-5p and interaction with DDX5, respectively, in colorectal cancer [8,9]. Likewise, the LncRNA TOB1-AS1 modulates the activation of the Wnt/β-catenin pathway by negatively regulating miR-23a in gastric cancer [10].

The HOX transcription antisense intergenic RNA (HOTAIR), has been described as a positive promoter of the PI3K/AKT and Wnt/β-catenin signaling pathways in several types of cancer, even in the presence of drugs and specific inhibitors for these pathways [11,12]. In ovarian and breast cancer, for example, HOTAIR overexpression has been linked to resistance to cisplatin and doxorubicin through activation of the Wnt/β-catenin and PI3K/AKT pathways, by favoring overexpression of β-catenin and maintaining sustained signaling of PI3K/AKT/mTOR pathway effectors, respectively [13,14]. Furthermore, HOTAIR modulates the activation of both signaling pathways in the context of cisplatin resistance by down-regulating miR-34a in gastric cancer [15].

The PI3K/AKT signaling pathway is activated through the interaction of a growth factor, cytokines or hormones with its tyrosine kinase receptor, which allows the PI3K protein embedded in the plasma membrane to be recruited and activated by phosphorylation of its catalytic subunit p110, allowing the conversion of phosphatidyl inositol (4,5)-bisphosphate (PIP2) to phosphatidyl inositol triphosphate (PIP3) [16]. During this process, the negative regulation of signaling is mediated by Phosphatase and tensin homolog (PTEN), which negatively regulates PI3K by dephosphorylating PIP3 and favoring the presence of PIP2. However, in the CC molecular context, the expression of PTEN is downregulated mainly by the methylation of its promoter [17]. Therefore, negative regulation of AKT is attenuated, allowing it to phosphorylate downstream mTORC1 and thereby regulate an increase in HIF1α at the protein level through the cooperation of mTOR effectors 4E-BP1 and S6 [18,19].

Additionally, it has been noted that HIF1α can induce abnormal activation of nuclear β-catenin in colorectal cancer when exposed to 5-Fluorouracil. The interaction between HIF1α and β-catenin is mediated by transcriptional effectors like BCL9 in hepatocarcinoma cells and TCF4 in glioblastoma, ensuring transcriptional equilibrium on specific targets [20,21,22].

The intricate relationship between the PI3K/AKT and Wnt/β-catenin signaling pathways has resulted in their consideration as a unified pathway in clinical treatment analyses. Inhibition of one pathway often leads to the inactivation of the other [23,24]. This synergy is particularly evident in studies involving colorectal cancer, where high levels of nuclear β-catenin correlate with resistance to PI3K/AKT pathway inhibitors. Effective outcomes are typically observed only with combined treatment strategies involving PI3K/AKT inhibitors (such as API2 and BKM120) alongside Wnt/tankyrase inhibitors (like NVP-TNKS656 and XAV-939) [25,26]. Similarly, the synergy of Wnt inhibitors directed against PORCN protein (ETC-159) and PI3K inhibitors (GDC-0941) has been observed in in vitro and in vivo models of pancreatic cancer [27].

Although the positive regulation of the Wnt/β-catenin and PI3K/AKT signaling pathways by HOTAIR has been well-documented in various types of cancer, the mechanisms interconnecting these pathways remain largely unexplored. This study reveals that HOTAIR orchestrates the transcriptional inactivation of PTEN by promoting hypermethylation of its promoter region. This epigenetic modification is facilitated by the recruitment of the enzyme DNMT1 by HOTAIR, thereby regulating both signaling pathways.

## 2. Materials and Methods

### 2.1. Cell Culture

Commercial cervical cell lines HeLa, SiHa, and HaCat were obtained from the Functional Genomics Laboratory of the National Cancer Institute of México. These cell lines were cultured in 100 × mm Petri dishes using Dulbecco’s Modified Eagle’s Medium (DMEM) High Glucose medium (Gibco, New York, NY, USA), supplemented with 10% Fetal Bovine Serum (FBS) (Corning, New York, NY, USA). Incubation was carried out at 37 °C in a 5% CO_2_ atmosphere.

### 2.2. Pathway Inhibitors

The inhibitors used for the Wnt/β-catenin pathway included ICRT14 (Toronto Research Chemical Canada I163900) and C59 (Bio-vision 2063-5, San Francisco, CA, USA), while the PI3K/AKT pathway was inhibited using Buparlisip (BKM120; NVP-BKM120, HY-70063, Sigma-Aldrich, St. Louis, MO, USA) solubilized in dimethyl sulfoxide (DMSO; Sigma-Aldrich, St. Louis, MO, USA). After 24 h of incubation with the IC50 of each inhibitor, the appropriate experiments were conducted.

### 2.3. HOTAIR Knockdown and Overexpression

To inhibit HOTAIR expression, HOTAIR ASO (antisense oligonucleotide sequence, HOTAIR-1084 2OMe/PS) [28] was utilized at concentrations ranging from 10 to 200 nM. For HOTAIR overexpression, a pcDNA 3.1 HOTAIR construct from GeneArt Thermo was used at a concentration of 1.25 μg/μL. Transfections were conducted using Lipofectamine 3000 (Invitrogen, Waltham, MA, USA) as per the manufacturer’s instructions. RNA extraction was performed 24 h post-transfection using Trizol (Invitrogen, Thermo-Fisher, Waltham, MA, USA).

### 2.4. MTT Assay

The IC50 values for the inhibitors were determined using an MTT assay. Cells (4000 per well) were seeded in a 96-well plate and treated with specific concentrations of inhibitors dissolved in 0.01% dimethyl sulfoxide (DMSO, Sigma-Aldrich St. Louis, MO, USA). After 24 h of incubation, the cells were treated with medium plus MTT added for colorimetric analysis.

### 2.5. TOP Flash Assay

To assess Wnt/β-catenin pathway activity, cells (500,000 per well) were seeded in 6-well plates and transfected with TOP and FOP flash vectors. Then, 24 h post-transfection, luciferase assays were performed following manufacturer protocols (Promega) by transfecting 2.5 μg of each plasmid.

### 2.6. HIF1α Transcriptional Activity Luciferase Assay

The evaluation of HIF1α transcriptional activity utilized the pGL4.42 [luc2P/HRE/Hygro] vector. Cells (100,000 per well) were seeded in 24-well plates and co-transfected with PGL 4.42 and pRL-null vector (Promega, Madison, WI, USA) for normalization. Then, 24 h post-transfection, luciferase assays were performed following manufacturer protocols (Promega, Madison, WI, USA), by transfecting 1 μg and 1 ng of each plasmid, respectively.

### 2.7. Tissue Expression for In Silico Meta-Analysis

Protein expression of PI3K, AKT, HIF1α, GSK3β, β-catenin, DNTM1, 3A, 3B, and PTEN was assessed using antibody-based proteomic data from The Human Protein Atlas, comparing staining intensity between healthy (3 samples) and cancerous tissues (13 samples).

### 2.8. In Silico Analysis

Pathway enrichment analysis related to HOTAIR was performed, predicting interactions with RNA-binding proteins (RBPs) using RBPDB (105 proteins) and Encori (60 proteins) databases (Appendix A). Further pathway and transcription factor enrichment analysis was conducted using WebGestalt and ShinyGo (version 2024) databases.

### 2.9. Western Blot

In order to evaluate the PI3K/AKT pathways, Western blot experiments were performed. For this purpose, 400,000 cells were seeded in 6-well plates. After treatment with BKM120 inhibitor (24 h), protein extraction was performed using RIPA lysis buffer (Santa Cruz Biotechnology, Dallas, TX, USA). For immunodetection, 50 μg of protein was used and electrophoresed on a 10% SDS-PAGE and transferred to a PVDF membrane (Amersham-GE Healthcare, Chalfont, UK). Finally Western blots were performed for respective components of the PI3K/AKT signaling pathway, PI3K (Cell signal. 4255S) and p-PI3K (BioqSs, bs-3332R), AKT (Cell signal. 9272S), and p-AKT s473 (Cell signal. 9271S) in 1:1000 dilutions. As a reference protein, β-actin (Santa Cruz, H1721) was used. Immunoreactivity was detected using the C-DiGit Blot Scanner kit (LI-COR, Lincoln, NE, USA).

### 2.10. Fluorescent In Situ Hybridization (FISH)

For the detection of HOTAIR using in situ hybridization, 60,000 cells were seeded into 12-well plates. The cells were fixed with 4% paraformaldehyde for 30 min and then hybridized with the HOTAIR Stellaris FISH Probes (Human HOTAIR CAL Fluor RED 590 Dye, VSMF-2176–5) at a 1:1000 dilution for 2 h at 37 °C. Subsequently, the nuclei were stained with Hoechst at a 1:3000 dilution (YB2820692). Images were captured using confocal microscopy provided by the National Cancer Institute and analyzed with Imaris Viewer software (9.0) (Bitplane) provided by Project 302978 to Michael Schnoor at Cinvestav-IPN.

### 2.11. qPCR

Oligos designed for qPCR evaluated mRNA expression of Wnt/β-catenin pathway target genes (Cyclin D1, c-Myc), HIF1α target gene (Glut1, HK2), HOTAIR, PTEN (mRNAs and Promoters) and DNMT1 (all alignment conditions are included in Appendix A) using the NCBI Gen database for design. Finally, we used GoTaq Green Mastermix for PCR and SYBR Green Master Mix (Thermo-Fisher, Waltham, MA, USA) for real-time PCR, following manufacturer protocols.

### 2.12. In Silico Analysis of DNA Methylation to PTEN, Expression, and Clinical Data

DNA methylation patterns in the PTEN promoter were analyzed using MEXPRESS [28] and Genome Browser tools, alongside CG sequence analysis in MethPrimer and Eukaryotic Promoter Data Base (EDP).

### 2.13. DNA Methylation Immunoprecipitation (IP 5mC)

DNA methylation immunoprecipitation was performed using Methylated-DNA IP Kit (Zymo, D55101-A, Irvine, CA, USA), according to the manufacturer’s instructions. The DNA was sonicated for 30 s and four pulses. Finally, PTEN promoter (Appendix A) was amplified by RT-PCR by standard conditions and then analyzed by electrophoresis on 1.5% agarose gel.

### 2.14. Chromatin Immunoprecipitation (ChIP) Assay

ChIP assays utilized EZ-ChIP KIT (Millipore) with 1,000,000 cells per dish, crosslinking DNA and protein. DNA was sonicated for 30 s and four pulses. Immunoprecipitating with antibodies against HIF1α was performed using 5 μg of antibody (Invitrogen, PA116601). HOTAIR promoter (Appendix A) was amplified by RT-PCR by standard conditions and then analyzed by electrophoresis on 1.5% agarose gel.

### 2.15. RNA Binding Protein Immunoprecipitation (RIP) Assay

Interaction probability between HOTAIR and transcriptional effectors (β-catenin and HIF1α) was analyzed using RPIseq. RNA binding protein immunoprecipitation (RIP) with specific antibodies to DNMT1 (abcam, Ab92314), β-catenin (abcam, ab22656) and HIF1α (Invitrogen, PA116601) was conducted using the Magna RIP RNA-binding protein immunoprecipitation kit (17–704, EMD Millipore, Burlington, MA, USA). For this assay, HeLa cell line was lysed by RIP lysis buffer, after which the lysate was incubated on Sepharose beads associated to protein A, conjugated with 5 μg to each antibody and negative control normal rabbit IgG (Millipore). Samples were incubated with proteinase K, and RNA was isolated. Finally, we performed RT-qPCR to corroborate HOTAIR association as mentioned before.

### 2.16. Statistical Analysis

Data are presented as mean ± standard deviation from three independent experiments. Statistical significance was assessed using one-way ANOVA or Student’s *t*-test (* *p* < 0.05, ** *p* < 0.01, *** *p* < 0.001).

## 3. Results

### 3.1. HOTAIR Associates with the Transcriptional Effectors β-Catenin and HIF1α

Our literature review and background research identified LncRNA HOTAIR as a crucial modulator of the Wnt/β-catenin and PI3K/AKT pathways in cervical cancer (CC), even in the presence of Wnt inhibitors. To validate this, we performed an in silico analysis to investigate whether HOTAIR interacts with these signaling pathways and their key transcriptional effectors, β-catenin and HIF1α.

Enrichment analyses focusing on cancer signaling pathways and transcription factors demonstrated a significant association of HOTAIR with the transcriptional regulation of both pathways (Figure 1A,B). Further examination of HOTAIR expression in CC cell lines showed consistent results, with HOTAIR notably overexpressed in the HeLa cell line compared to non-tumor cell lines (Figure 1C).

To delve deeper into the potential interaction of HOTAIR with β-catenin and HIF1α, we conducted both in silico assessments and experimental validation using an RIP assay in the HeLa cell line. Our results revealed significant enrichment in the association between HOTAIR and these transcription factors (Figure 1D), suggesting that HOTAIR could act as a central regulatory hub for both signaling pathways.

### 3.2. PI3K/AKT Regulates HOTAIR Expression and Wnt/β-Catenin Transcriptional Activity in Cervical Cancer

After establishing HOTAIR’s role in modulating key signaling pathways in cervical cancer (CC), we conducted a meta-analysis that revealed elevated levels of PI3K, AKT, HIF1α, and β-catenin proteins in CC tumor samples compared to healthy controls, alongside reduced levels of GSK3β (Appendix A). Consistent with these findings, we investigated the effects of the Wnt/β-catenin pathway inhibitors C59 and ICRT14 on transcriptional activity and the PI3K/AKT pathway in different CC cell lines. We observed a specific reduction in PI3K pathway activation in the SiHa cell line, but not in the HeLa cell line (Appendix A). These results, combined with those from Figure 1, suggest that the regulatory mechanisms of HOTAIR are cell-type specific and depend on the molecular context of each cell line.

Subsequently, we determined the half-maximal inhibitory concentration (IC50) of BKM120 and confirmed its efficacy in both cell lines through Western blotting, luciferase assays for HIF1α activity, and qPCR analysis of Glut1 and Hexokinase 2 (HK2) (Figure 2A,B, Appendix A). Using a FISH assay, we detected HOTAIR with a uniform distribution in both the nucleus and cytoplasm under basal conditions. However, treatment with HOTAIR Antisense Oligonucleotide (ASO HOTAIR) and BKM120 resulted in a significant reduction in HOTAIR levels compared to controls treated with DMSO, scramble control, or left untreated. This suggests that the PI3K/AKT pathway may regulate HOTAIR expression. This regulation was further supported by qPCR analysis, which demonstrated a significant decrease in HOTAIR expression following BKM120 treatment in HeLa cells (Figure 2C,D). Interestingly, in the SiHa cell line, BKM120 treatment led to an increase in HOTAIR expression (see Appendix A), indicating a possible compensatory mechanism that differs between the two cell lines.

We further investigated whether PI3K/AKT pathway inhibition affects Wnt/β-catenin transcriptional activity. In line with the inhibition of HOTAIR, we observed a decrease in Wnt pathway transcriptional activity in HeLa cells, evidenced by reduced expression of Cyclin D1 and c-Myc, key transcriptional targets of this pathway (Figure 2D). These findings suggest that the PI3K/AKT pathway sustains Wnt/β-catenin transcriptional activity through the regulation of HOTAIR expression.

### 3.3. HOTAIR Regulates Wnt/β-Catenin and PI3K/AKT Transcriptional Activity in Cervical Cancer

HOTAIR regulates the activation of PI3K/AKT and Wnt/βcatenin pathways in CC. Once the association between HOTAIR and the transcriptional effectors β-catenin and HIF1α was corroborated, both knockdown and overexpression of HOTAIR were performed on the HeLa cell line (Figure 3A,B) and only overexpression on the SiHa cell line (Appendix A). Subsequently, we evaluated the activation of the PI3K/AKT pathway and the transcriptional activity of HIF1α upon inhibition of HOTAIR, which resulted in a decrease in HIF1α transcriptional activity (Figure 3C).

Conversely, overexpression of HOTAIR led to an increase in HIF1α transcriptional activity in both cell lines, as corroborated through its transcriptional targets Glut1 and HK2 (Figure 3D, Appendix A). Additionally, we assessed the transcriptional activity of the Wnt/β-catenin signaling pathway following HOTAIR knockdown, observing a decrease in its transcription activity (Figure 3E). Conversely, overexpression of HOTAIR resulted in an increase in Wnt/β-catenin signaling pathway activity in both cell lines, as corroborated through its transcriptional targets Cyclin D1 and c-Myc (Figure 3F, Appendix A). These findings collectively indicate that HOTAIR regulates the activation of both the PI3K/AKT and Wnt/β-catenin signaling pathways in the HeLa and SiHa cell lines.

### 3.4. HOTAIR Expression Is Regulated by a Feedback Mechanism through HIF1α

Several studies suggest that LncRNAs like HOTAIR may undergo regulation through feedback loops involving signaling pathways and the lncRNA itself. Therefore, we investigated whether the transcriptional regulation of HOTAIR involves a feedback loop mechanism with the PI3K/AKT and Wnt/β-catenin pathways. Initially, we analyzed the HOTAIR promoter sequence using the Genome Browser, identifying multiple Wnt response elements (WRE) and hypoxia response elements (HRE). This finding suggested potential transcriptional regulation mediated by transcriptional effectors of both pathways (Figure 4A). Subsequently, to examine if HIF1α transcriptionally activates HOTAIR, we stabilized HIF1α using the prolyl hydroxylase inhibitor DMOG at concentrations ranging from 100μM to 1mM. The effectiveness of HIF1α stabilization was confirmed by assessing its transcriptional activity (Figure 4B,C).

Similarly, we induced over-activation of the Wnt/β-catenin pathway by treating HeLa cells with 25 mM LiCl to stimulate transcription (Figure 4D). Following the stabilization of these signaling pathways, we assessed HOTAIR expression under conditions of transcriptional over-activation of both pathways. Interestingly, we observed an increase in HOTAIR expression when HIF1α was stabilized, but not when the Wnt/β-catenin pathway was over-activated alone (Figure 4E,F). To confirm the direct involvement of HIF1α in the transcriptional regulation of HOTAIR, we performed a chromatin immunoprecipitation (ChIP) assay targeting HIF1α, which showed HIF1α binding to the HOTAIR promoter region (Figure 4G). This ChIP result strongly suggests that HIF1α transcriptionally activates HOTAIR through a feedback loop mechanism.

### 3.5. HOTAIR Regulates the PI3K/AKT Pathway by Repressing PTEN

To further elucidate the regulatory role of HOTAIR in maintaining pathway activation, we investigated the potential transcriptional repression of PTEN, a critical phosphatase antagonist in the PI3K/AKT pathway. We utilized the MEXPRESS database to analyze PTEN promoter methylation patterns across 317 samples, and identified potential methylation sites within the PTEN promoter region using the Genome Browser and MethPrimer tools. This analysis revealed a significant prevalence of PTEN promoter methylation (Appendix A). Moreover, an in silico meta-analysis indicated reduced PTEN expression levels, which were correlated with elevated expression of DNMT1, DNMT3A, and DNMT3B (Appendix A). These findings suggest that HOTAIR may facilitate the methylation of the PTEN promoter through the recruitment of DNA methyltransferase enzymes, thereby contributing to the repression of PTEN expression and the activation of downstream pathways.

We analyzed PTEN expression levels in the HeLa cell line and observed a decrease compared to non-tumoral cell lines (Figure 5A). To investigate whether HOTAIR influences PTEN expression, we performed knockdown and overexpression experiments. Remarkably, inhibiting HOTAIR led to an increase in PTEN expression, while overexpressing HOTAIR resulted in a decrease in PTEN levels at both the RNA and protein levels (Figure 5B). These results indicate that PTEN expression in HeLa cells is dependent on HOTAIR.

On the basis of previous results and the detection of DNMT1 in patient specimens (Appendix A), we assessed DNMT1 expression levels in the HeLa cells. Concordantly, DNMT1 is overexpressed in tumor cells compared to non-tumoral cell lines (Figure 5C). We confirmed by RIP assay the potential interaction between DNMT1 and HOTAIR (Figure 5D). This suggests that HOTAIR may regulates PTEN expression through DNMT1-mediated mechanisms.

Finally, we performed an immunoprecipitation (IP) assay targeting 5-methylcytosine (5-mC) to investigate whether HOTAIR influences the methylation pattern of the PTEN promoter. These results indicated that HOTAIR knockdown led to a decrease in 5-mC levels, while HOTAIR overexpression increased 5-mC levels (Figure 5E). This strongly suggests that HOTAIR can regulate partial methylation of the PTEN promoter, thereby promoting its downregulation.

## 4. Discussion

As previously noted, the regulation between the PI3K/AKT and Wnt/β-catenin pathways is crucial for the development of cervical cancer (CC). Our initial analysis, based on data from the Protein Atlas, revealed a higher presence of PI3K/AKT pathway components in cervical cancer samples compared to healthy tissues. This observation aligns with the existing literature, which reports over-activation of this pathway in up to 31% of squamous cell carcinomas and 24% of CC adenocarcinomas. Additionally, mutations in the PIK3CA gene, which promote PI3K overexpression, have been identified in up to 36% of CC cases [29]. Inhibition of the PI3K/AKT pathway using the inhibitor BKM120, known for its efficacy in various cancers including esophageal carcinoma, breast cancer, glioblastoma, and neuroendocrine cervical carcinoma cell lines, was subsequently explored [30,31].

We observed that inhibiting the PI3K/AKT pathway led to the transcriptional inactivation of the Wnt/β-catenin pathway, suggesting a positive regulatory relationship between these pathways in HeLa and SiHa cell lines. This mechanism is supported by previous reports, indicating that AKT-mediated inactivation of GSK3β through phosphorylation (Ser9, 21) and mTORC1’s inhibition of autophagy could facilitate Wnt/β-catenin pathway activation [32,33].

Further analysis included the detection of Wnt/β-catenin pathway components in CC samples compared to healthy tissues, followed by pathway inactivation using the inhibitors C59 and ICRT14 in HeLa and SiHa cell lines. Consistent with Protein Atlas data, prior studies reported β-catenin overexpression in up to 67% and GSK3β inactivation in up to 56% of CC cases. We found that inhibition of the Wnt/β-catenin pathway resulted in the inhibition of the PI3K/AKT pathway in the SiHa cell line, but not in HeLa cells. This differential response may indicate that in SiHa cells, GSK3β inactivation could enhance mTORC1 activation and other transcriptional effectors, maintaining a positive feedback loop between pathways [34]. However, the mechanisms sustaining pathway interaction in HeLa cells remain unclear, suggesting the involvement of additional intermediaries, such as the long non-coding RNA (lncRNA) HOTAIR.

Given these observations, we investigated HOTAIR’s role in modulating the Wnt/β-catenin and PI3K/AKT pathways. Our results demonstrated that HOTAIR is crucial for the activation of both pathways, influencing the transcriptional activity and expression of key targets. These findings align with previous research linking HOTAIR to Wnt/β-catenin pathway modulation in leukemia and other cancers [13,14,35,36]. HOTAIR has also been implicated in maintaining Wnt/β-catenin pathway activation through methylation processes in CC [37]. Additionally, HOTAIR has been associated with PI3K/AKT pathway regulation in breast cancer, retinoblastoma, and gastric cancer via microRNA regulation [13,38,39]. In CC, while HOTAIR’s role is less established, it has been reported to regulate migration via miR-29b and PI3K/AKT activation [40].

Further, we explored whether HOTAIR-mediated regulation involved a feedback loop with these signaling pathways. We identified WRE and HRE in the HOTAIR promoter sequence, demonstrating that transcriptional regulation is mediated solely by HIF1α, consistent with reports indicating HIF1α regulation of HOTAIR under hypoxic conditions [41]. This suggests that PI3K/AKT pathway’s positive regulation of the Wnt/β-catenin pathway in HeLa cells may be facilitated through HOTAIR expression.

The specificity of these mechanisms in HeLa, and not in SiHa cells, suggests that aberrant PI3K/AKT pathway activation may involve P53 mutations, leading to PTEN instability and HOTAIR overexpression as a central regulatory molecule. In CC, P53 regulation of PI3K is critical for PTEN-dependent or independent tumor progression [42,43]. Thus, P53 mutations in HeLa cells could result in HOTAIR overexpression and its associated oncogenic mechanisms. Recent studies have demonstrated the translational potential of inhibiting HOTAIR through molecular docking with two inhibitors, including AC1NOD4Q. This inhibitor targets the 5’ domain responsible for interaction with EZH2, resulting in the inhibition of metastasis in an orthotopic breast cancer model using cell lines [44]. Notably, these findings were replicated in patient-derived xenograft models, underscoring the feasibility of this approach. In addition, the potential role of the lncRNAs as biomarkers of prognosis, treatment response and therapeutic targets is becoming increasingly important. Currently, more than 39 international clinical trials are ongoing, mainly in phases I and II. In particular, HOTAIR has been registered as a biomarker in thyroid cancer. In a study involving 89 patients, HOTAIR was found to be significantly overexpressed (ΔCt value: 5.74 ± 1.129 versus 8.22 ± 1.301, *p* < 0.001), underlining its potential utility as a clinical biomarker [45]. Furthermore, several authors have associated high HOTAIR levels with poor clinical outcomes, metastasis development, therapy response, recurrence and bad overall survival of patients with breast, gastric, uterine, and hepatocellular carcinomas [46,47,48,49]. Although HIF-1 and PI3K pharmacological inhibition showed benefits in the inhibition of angiogenesis, invasiveness, proliferation, de-differentiation, and tumor growth in several solid tumors, their clinical use has demonstrated poor effectiveness due to low specificity and toxic side effects [30,50]. Therefore, the focus on using RNA interference molecules or small molecules designed through molecular docking that targets HOTAIR to modulate the PI3K/AKT/ HIF1a axis could be a further option to avoid chemoprevention.

To conclude, we examined how HOTAIR regulates PTEN expression. We observed decreased PTEN mRNA and protein levels in CC samples and increased PTEN expression upon HOTAIR inhibition in HeLa cells. These findings corroborate previous studies reporting PTEN loss in up to 62% of CC cases, leading to sustained PI3K/AKT pathway activation [51,52]. HOTAIR, interacting with DNMT1, may downregulate PTEN through methylation, suggesting a feedback loop involving HIF1α/HOTAIR/PTEN that activates the Wnt/β-catenin pathway [53,54,55].

## 5. Conclusions

This study reveals a novel regulatory mechanism by which HOTAIR mediates positive crosstalk between the Wnt/β-catenin and PI3K/AKT signaling pathways in CC. The mechanism involves a feedback loop wherein HIF1α induces HOTAIR overexpression, which in turn sustains transcriptional activity in the Wnt/β-catenin pathway and possibly enables continued PI3K/AKT pathway activation by methylating the PTEN promoter. This complex regulatory network underscores HOTAIR’s pivotal role in coordinating these pathways and its significance in cervical cancer progression (Figure 6). This critical regulatory role suggests that HOTAIR could serve as a promising therapeutic target in cancers characterized by PTEN downregulation.

## Figures and Tables

**Figure 1 cells-13-01484-f001:**
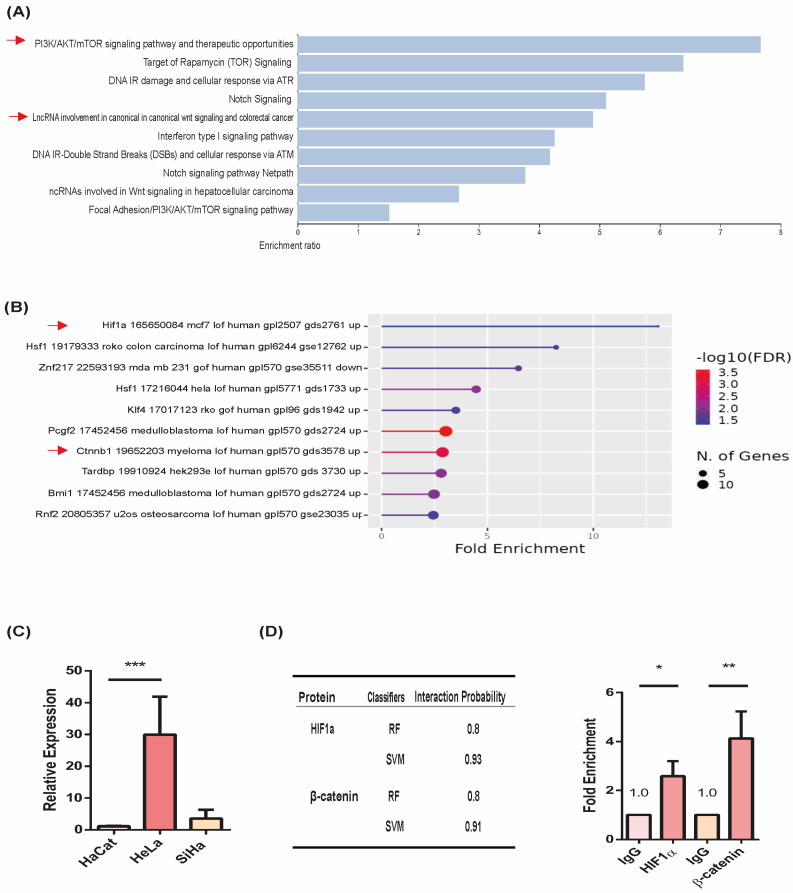
Enrichment of HOTAIR-associated transcription factors and pathways. (**A**) Enrichment of HOTAIR-associated signaling pathways from Webgestalt. (**B**) Enrichment of transcription factors associated with HOTAIR from Shinygo. (**C**) HOTAIR expression in CC cell lines in comparison with non-tumoral cell line. (**D**) Interaction probability of HOTAIR with β-catenin and HIF1α and its experimental corroboration by RIP assay. *p* < 0.05 (*). *p* < 0.01 (**). *p* < 0.001 (***).

**Figure 2 cells-13-01484-f002:**
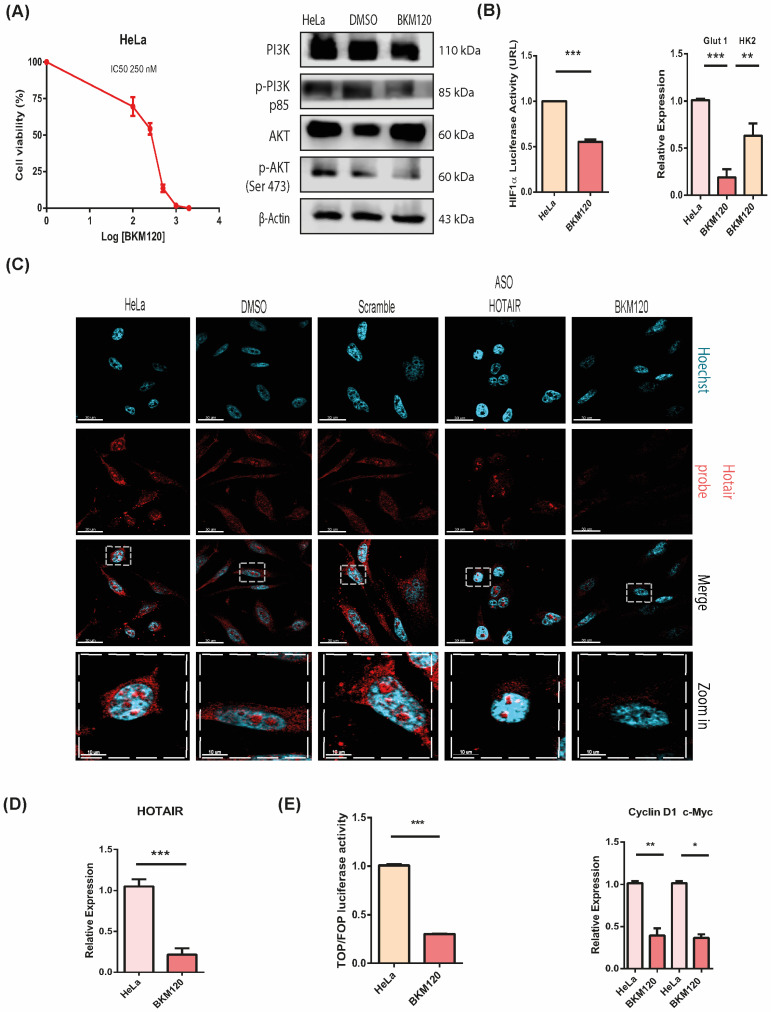
PI3K/AKT regulates HOTAIR expression and Wnt/β-catenin activation in HeLa cell line. (**A**) IC50 and Western blot of PI3K/AKT pathway with BKM120 inhibitor. (**B**) BKM120 inhibits transcriptional activity of HIF1α, determined by measuring luciferase reporter and Glut1 and HK2 expression in HeLa cells. (**C**) HOTAIR levels detected in HeLa cells treated with anti-sense probe and BKM120 inhibitor. Bar = 30 and 10 μm. (**D**) Inhibition of HOTAIR expression with BKM inhibitor treatment. (**E**) Wnt/β-catenin transcriptional activity evaluated by TOP Flash activity and Cyclin D1, c-Myc expression with BKM120 inhibitor treatment. *p* < 0.05 (*). *p* < 0.01 (**). *p* < 0.001 (***).

**Figure 3 cells-13-01484-f003:**
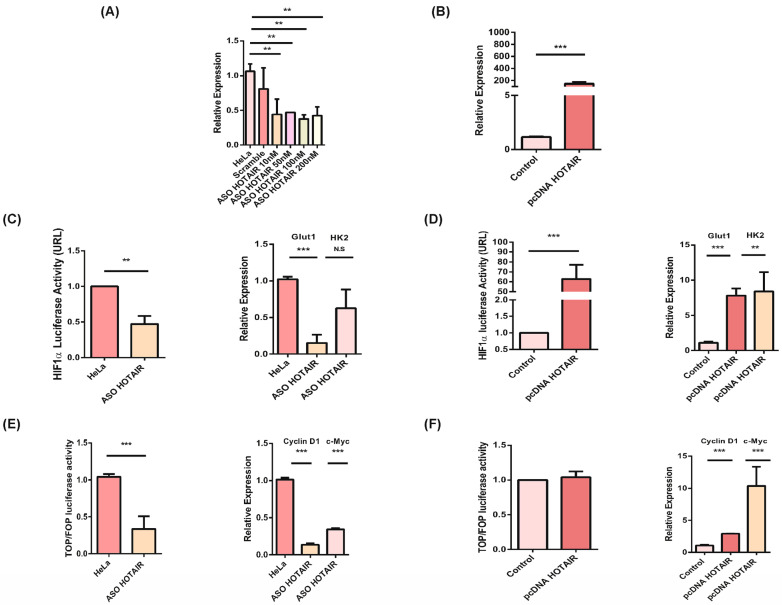
HOTAIR regulates the Wnt/β-catenin and PI3K/AKT pathways in the HeLa cell line. (**A**) Expression levels of HOTAIR by different amounts of anti-sense probe. (**B**) HOTAIR overexpression in HeLa cell line. (**C**) Inhibition of HOTAIR affects transcriptional activity of HIF1α by luciferase activity, and HIF1α targets Glut1 and HK2 expression. (**D**) HOTAIR overexpression increases HIF1α transcriptional activity by luciferase activity, and HIF1α targets expression. (**E**) HOTAIR knockdown decreases Wnt/β-catenin transcriptional activity, as evaluated by TOP Flash activity and Cyclin D1, c-Myc expression. (**F**) HOTAIR overexpression increases Wnt/β-catenin transcriptional activity, as evaluated by TOP Flash activity and Cyclin D1, c-Myc. *p* < 0.05 (*). *p* < 0.01 (**). *p* < 0.001 (***).

**Figure 4 cells-13-01484-f004:**
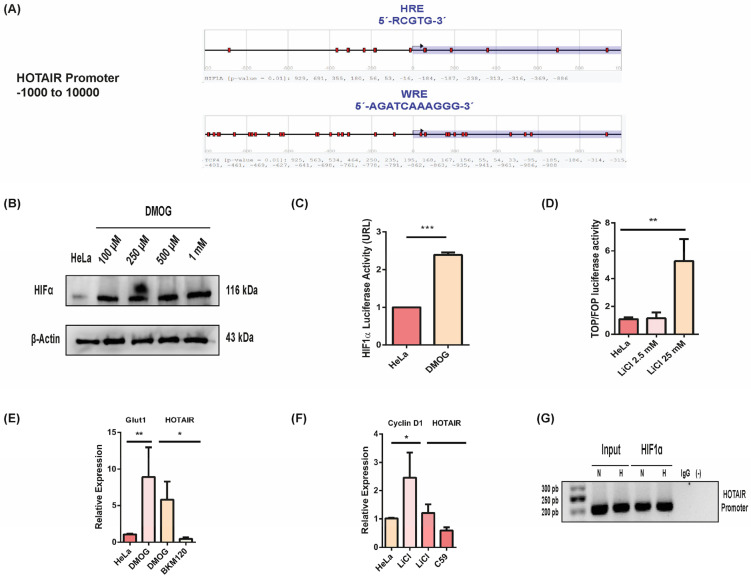
HIF1α-mediated expression of HOTAIR in the HeLa cell line (**A**) HOTAIR promoter is enriched with Wnt response elements (WRE) and hypoxia response elements (HRE). (**B,C**) Stabilization of HIF1α by DMOG and assessment of its transcriptional activity in the HeLa cell line. (**D**) Over-activation of the transcriptional activity of Wnt/β- catenin by LiCl in HeLa cell line. (**E,F**) HOTAIR expression upon over-activation of HIF1α and Wnt/β-catenin pathway transcriptional activity by LiCl and DMOG in HeLa cell line. (**G**) HIF-1α is located in HOTAIR promoter by ChIP assay. *p* < 0.05 (*). *p* < 0.01 (**). *p* < 0.001 (***).

**Figure 5 cells-13-01484-f005:**
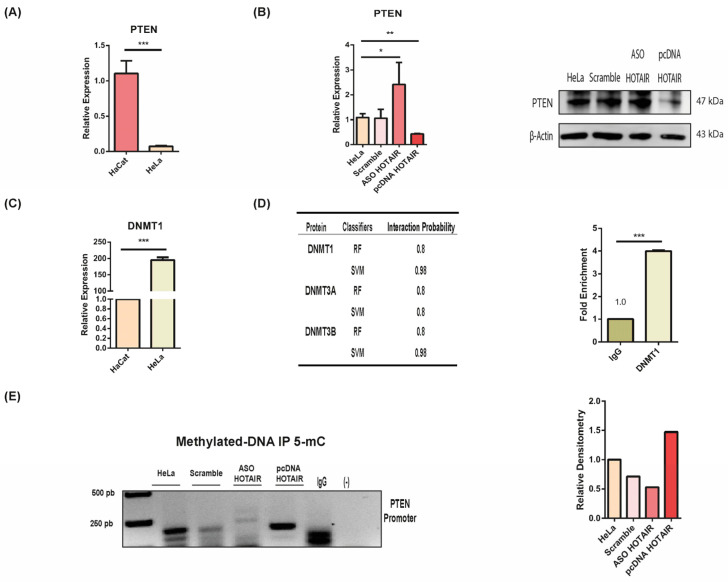
HOTAIR-mediated expression of PTEN. (**A**) PTEN expression levels in HeLa cell line. (**B**) PTEN expression upon HOTAIR inhibition and overexpression evaluated by qPCR and Western blot. (**C**) DNMT1 expression levels in HeLa cell line. (**D**) Interaction probability of HOTAIR with DNMT1 and its experimental corroboration by RIP assay. (**E**) 5-mC IP of PTEN promoter upon HOTAIR inhibition and overexpression. *p* < 0.05 (*). *p* < 0.01 (**). *p* < 0.001 (***).

**Figure 6 cells-13-01484-f006:**
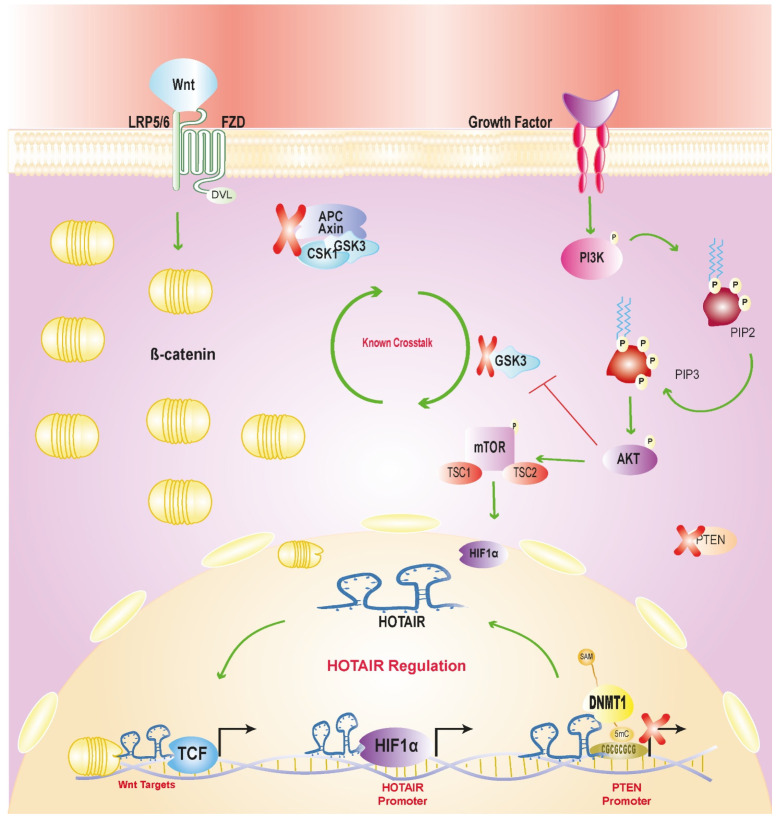
HOTAIR-mediated over-activation of PI3K/AKT and Wnt/β-catenin signaling pathways through nuclear processes, methylation of PTEN promoter and feedback with HIF1α.

## Data Availability

Any data will be distributed upon request to carlos.pplas@gmail.com and samuel.trujano1@gmail.com.

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
