# Peer review of "HOTAIR Promotes the Hyperactivation of PI3K/Akt and Wnt/β-Catenin Signaling Pathways via PTEN Hypermethylation in Cervical Cancer"

_cells, 2024, doi:10.3390/cells13171484_

Round 1
Reviewer 1 Report
Comments and Suggestions for Authors
In the article titled “HOTAIR drives the overactivation of PI3K/Akt and Wnt/β-catenin signaling pathways via PTEN hypermethylation in cervical cancer”, Trujano-Camacho et al. investigated the role of a long non-coding RNA, HOTAIR (HOX transcription antisense intergenic RNA), in cervical cancer. Specifically, they showed that HOTAIR mediates the activation of the PI3K/AKT and Wnt/β-catenin pathways through a feedback-loop mechanism in which HIF1α favors HOTAIR overexpression which, in turn, promotes both the transcriptional activity of the Wnt/β-catenin pathway and supports the activation of the PI3K/AKT pathway through PTEN down-regulation by the methylation of its promoter.
Overall, this is an interesting study that could benefit cancer therapy. I have several suggestions that could further increase the impact of this work:
· The article is generally well-written, the language used is clear but there are some errors and repetitions:
-Line 14: “Instituto Nacional Nacional de Cancerología”. The word “Nacional” is repeated twice.
-Line 50: “has been extensively explored as essential effectors to signaling pathways”. “Effectors to” could be changed to “effectors of”.
-Line 56: “thereby promoting the progression of cancers such Cervical Cancer (CC)”. It would be preferred “such as Cervical Cancer (CC)”.
-Line 82: “During this process, the negative regulation of signaling is mediated by the phosphatase Phosphatase and tensin homolog (PTEN)”. The word “Phosphatase” is repeated twice;
-Line 87: “regulation of AKT is attenuated; allowing it to phosphorylate (..)”. Delete the semicolon “;”.
-Line 137: “Luciferase assays were performed following manufacturer protocols (Promega) transfected 2.5µg of each plasmid”. “Transfected” could be changed to “by transfecting”;
-Line 156: “For this purpose, 400 000 cells were seeded in 6-well plates (...)”. 400,000 is correct.
· The “Materials and Methods” section is generally well-described, but some details are missing e.g. those concerning Western Blot, qPCR.
· Subparagraph 3.2 “PI3K/AKT Regulates HOTAIR Expression and Wnt/β-catenin Transcriptional Activity in Cervical Cancer” and 3.3 “Regulation of HOTAIR Expression and Wnt/β-Catenin Transcriptional Activity by PI3K/AKT in Cervical Cancer” provide the same information and may be combined into one subparagraph.
· Some abbreviations are used without being defined initially (e.g., DMEM in the “Materials and Methods” section).
· The heading of the figure legends may be in bold;
· The article is well-structured but the text is a bit redundant and occasionally lacks specificity.
Comments on the Quality of English Language
The use of the English language is quite correct, some sentences could be rephrased more concisely but overall the text is well-written.
Author Response
Reviewer 1
In the article titled “HOTAIR drives the overactivation of PI3K/Akt and Wnt/β-catenin signaling pathways via PTEN hypermethylation in cervical cancer”, Trujano-Camacho et al. investigated the role of a long non-coding RNA, HOTAIR (HOX transcription antisense intergenic RNA), in cervical cancer. Specifically, they showed that HOTAIR mediates the activation of the PI3K/AKT and Wnt/β-catenin pathways through a feedback-loop mechanism in which HIF1α favors HOTAIR overexpression which, in turn, promotes both the transcriptional activity of the Wnt/β-catenin pathway and supports the activation of the PI3K/AKT pathway through PTEN down-regulation by the methylation of its promoter.
Overall, this is an interesting study that could benefit cancer therapy. I have several suggestions that could further increase the impact of this work:
- The article is generally well-written, the language used is clear but there are some errors and repetitions:
-Line 14: “Instituto Nacional Nacional de Cancerología”. The word “Nacional” is repeated twice.
- Dear reviewer, your comment was processed, and the duplicate word was removed.
-Line 50: “has been extensively explored as essential effectors to signaling pathways”. “Effectors to” could be changed to “effectors of”.
- Dear reviewer, your suggestion was taken into consideration and the corresponding adjustments were made on line 51.
-Line 56: “thereby promoting the progression of cancers such Cervical Cancer (CC)”. It would be preferred “such as Cervical Cancer (CC)”.
- The corresponding modification was made on line 56.
-Line 82: “During this process, the negative regulation of signaling is mediated by the phosphatase Phosphatase and tensin homolog (PTEN)”. The word “Phosphatase” is repeated twice;
- Dear reviewer, the word phosphatase belongs to the name of the PTEN protein, to avoid duplication of this word, the writing has been corrected on linea 83
-Line 87: “regulation of AKT is attenuated; allowing it to phosphorylate (..)”. Delete the semicolon “;”.
- The corresponding modification was made on line 87
-Line 137: “Luciferase assays were performed following manufacturer protocols (Promega) transfected 2.5µg of each plasmid”. “Transfected” could be changed to “by transfecting”;
- Dear reviewer, your suggestion was taken into consideration and the corresponding adjustments were made on line 138.
-Line 156: “For this purpose, 400 000 cells were seeded in 6-well plates (...)”. 400,000 is correct.
- The corresponding modification was made on line 161.
- The “Materials and Methods” section is generally well-described, but some details are missing e.g. those concerning Western Blot, qPCR.
- The corresponding modification was made in all section to Materials and Methods
- Subparagraph 3.2 “PI3K/AKT Regulates HOTAIR Expression and Wnt/β-catenin Transcriptional Activity in Cervical Cancer”and 3.3 “Regulation of HOTAIR Expression and Wnt/β-Catenin Transcriptional Activity by PI3K/AKT in Cervical Cancer” provide the same information and may be combined into one subparagraph.
- Dear reviewer, thanks to your comments we realized that the text in sub paragraph 3.3 was duplicated due to the fact that the correct text was not saved during the process of writing the manuscript. The proper text has been added and the title of the sub paragraph has been changed to make it clearer. These substantive amendments cover lines 273 to 299.
- Some abbreviations are used without being defined initially (e.g., DMEM in the “Materials and Methods” section).
- Dear reviewer, the definitions of the abbreviations have been added according to your suggestion.
- The headingof the figure legends may be in bold;
- The corresponding modification was made on the figures.
- The article is well-structured but the text is a bit redundant and occasionally lacks specificity.
- we work on the text in general to use more specific language
Reviewer 2 Report
Comments and Suggestions for Authors
Dear Colleagues,
Thank you for providing the opportunity to review this insightful manuscript on the role of the long non-coding RNA HOTAIR in regulating the PI3K/Akt and Wnt/β-catenin signaling pathways in cervical cancer. As a seasoned medical researcher with over three decades of experience, I commend your comprehensive approach and attention to detail in unraveling the intricate molecular mechanisms underlying this important clinical problem.
1. The primary objective of this study is to elucidate the pivotal role of HOTAIR in sustaining the activation of both the PI3K/Akt and Wnt/β-catenin signaling pathways in cervical cancer, with a specific focus on the regulatory axis involving HOTAIR and HIF1α.
2. The researchers have utilized a robust experimental methodology, including cell line models, pathway inhibitors, luciferase reporter assays, Western blotting, FISH, qPCR, ChIP, and RIP assays, to systematically investigate the complex interplay between these key signaling cascades. The in silico analyses further strengthened the study by providing valuable insights into the broader regulatory networks involving HOTAIR.
3. One potential limitation of the study is the reliance on a single cervical cancer cell line, HeLa, for the majority of the experimental work. It would be beneficial to validate the key findings across additional cell line models, preferably representing different subtypes or stages of cervical cancer, to enhance the generalizability of the conclusions.
4. Recommendations to further strengthen the manuscript:
a. Incorporate a more comprehensive discussion of the clinical relevance and potential translational implications of the findings.
b. Expand the discussion on the cell-type specific regulatory mechanisms observed between the HeLa and SiHa cell lines, as this could provide important insights into the heterogeneity of cervical cancer.
c. Consider including a graphical abstract or schematic to concisely summarize the proposed model and key signaling interactions.
5. Examples of potential language improvements:
a. "drives the overactivation of" could be rephrased as "promotes the hyperactivation of"
b. "HOTAIR is pivotal in maintaining the activation" could be revised to "HOTAIR is central in sustaining the activation"
c. "to the methylation of the PTEN promoter" could be modified to "to the epigenetic silencing of the PTEN promoter"
d. "positive crosstalk between the PI3K/AKT and Wnt/β-catenin pathways" could be expressed as "reciprocal activation of the PI3K/Akt and Wnt/β-catenin pathways"
e. "TOB1-AS1 regulates the activation of the Wnt/β-catenin pathway" could be improved to "TOB1-AS1 modulates the activation of the Wnt/β-catenin pathway"
6. Strengths of the study:
- Comprehensive and mechanistic approach to understanding the role of HOTAIR in cervical cancer pathogenesis
- Incorporation of both in vitro experiments and in silico analyses to provide a multi-layered understanding
Limitations:
- Reliance on a single cell line for the majority of the experimental work
- Lack of discussion on the potential clinical relevance and translational applications
7. To the authors, I commend your dedication to unraveling the complex regulatory networks governing cervical cancer progression. The insights provided in this manuscript represent a significant advancement in our understanding of the disease and open up promising avenues for future research and therapeutic development. Keep up the excellent work, and continue pushing the boundaries of our knowledge in this important field.
Comments on the Quality of English LanguageDear Colleagues,
Thank you for providing the opportunity to review this insightful manuscript on the role of the long non-coding RNA HOTAIR in regulating the PI3K/Akt and Wnt/β-catenin signaling pathways in cervical cancer. As a seasoned medical researcher with over three decades of experience, I commend your comprehensive approach and attention to detail in unraveling the intricate molecular mechanisms underlying this important clinical problem.
1. The primary objective of this study is to elucidate the pivotal role of HOTAIR in sustaining the activation of both the PI3K/Akt and Wnt/β-catenin signaling pathways in cervical cancer, with a specific focus on the regulatory axis involving HOTAIR and HIF1α.
2. The researchers have utilized a robust experimental methodology, including cell line models, pathway inhibitors, luciferase reporter assays, Western blotting, FISH, qPCR, ChIP, and RIP assays, to systematically investigate the complex interplay between these key signaling cascades. The in silico analyses further strengthened the study by providing valuable insights into the broader regulatory networks involving HOTAIR.
3. One potential limitation of the study is the reliance on a single cervical cancer cell line, HeLa, for the majority of the experimental work. It would be beneficial to validate the key findings across additional cell line models, preferably representing different subtypes or stages of cervical cancer, to enhance the generalizability of the conclusions.
4. Recommendations to further strengthen the manuscript:
a. Incorporate a more comprehensive discussion of the clinical relevance and potential translational implications of the findings.
b. Expand the discussion on the cell-type specific regulatory mechanisms observed between the HeLa and SiHa cell lines, as this could provide important insights into the heterogeneity of cervical cancer.
c. Consider including a graphical abstract or schematic to concisely summarize the proposed model and key signaling interactions.
5. Examples of potential language improvements:
a. "drives the overactivation of" could be rephrased as "promotes the hyperactivation of"
b. "HOTAIR is pivotal in maintaining the activation" could be revised to "HOTAIR is central in sustaining the activation"
c. "to the methylation of the PTEN promoter" could be modified to "to the epigenetic silencing of the PTEN promoter"
d. "positive crosstalk between the PI3K/AKT and Wnt/β-catenin pathways" could be expressed as "reciprocal activation of the PI3K/Akt and Wnt/β-catenin pathways"
e. "TOB1-AS1 regulates the activation of the Wnt/β-catenin pathway" could be improved to "TOB1-AS1 modulates the activation of the Wnt/β-catenin pathway"
6. Strengths of the study:
- Comprehensive and mechanistic approach to understanding the role of HOTAIR in cervical cancer pathogenesis
- Incorporation of both in vitro experiments and in silico analyses to provide a multi-layered understanding
Limitations:
- Reliance on a single cell line for the majority of the experimental work
- Lack of discussion on the potential clinical relevance and translational applications
7. To the authors, I commend your dedication to unraveling the complex regulatory networks governing cervical cancer progression. The insights provided in this manuscript represent a significant advancement in our understanding of the disease and open up promising avenues for future research and therapeutic development. Keep up the excellent work, and continue pushing the boundaries of our knowledge in this important field.
Author Response
Thank you for providing the opportunity to review this insightful manuscript on the role of the long non-coding RNA HOTAIR in regulating the PI3K/Akt and Wnt/β-catenin signaling pathways in cervical cancer. As a seasoned medical researcher with over three decades of experience, I commend your comprehensive approach and attention to detail in unraveling the intricate molecular mechanisms underlying this important clinical problem.
- The primary objective of this study is to elucidate the pivotal role of HOTAIR in sustaining the activation of both the PI3K/Akt and Wnt/β-catenin signaling pathways in cervical cancer, with a specific focus on the regulatory axis involving HOTAIR and HIF1α.
- The researchers have utilized a robust experimental methodology, including cell line models, pathway inhibitors, luciferase reporter assays, Western blotting, FISH, qPCR, ChIP, and RIP assays, to systematically investigate the complex interplay between these key signaling cascades. The in silico analyses further strengthened the study by providing valuable insights into the broader regulatory networks involving HOTAIR.
- One potential limitation of the study is the reliance on a single cervical cancer cell line, HeLa, for the majority of the experimental work. It would be beneficial to validate the key findings across additional cell line models, preferably representing different subtypes or stages of cervical cancer, to enhance the generalizability of the conclusions.
- Dear reviewer, thank you so much for your thoughtful and constructive comments.
- Recommendations to further strengthen the manuscript:
- Incorporate a more comprehensive discussion of the clinical relevance and potential translational implications of the findings.
- Expand the discussion on the cell-type specific regulatory mechanisms observed between the HeLa and SiHa cell lines, as this could provide important insights into the heterogeneity of cervical cancer.
- Consider including a graphical abstract or schematic to concisely summarize the proposed model and key signaling interactions.
- Dear reviewer, we made your improvements to our manuscript
- Examples of potential language improvements:
- "drives the overactivation of" could be rephrased as "promotes the hyperactivation of"
- Dear reviewer, thanks to your comments we realized the modifications according to your suggestion
- "HOTAIR is pivotal in maintaining the activation" could be revised to "HOTAIR is central in sustaining the activation"
- Dear reviewer, we made the modifications according to your suggestion on line 37
- "to the methylation of the PTEN promoter" could be modified to "to the epigenetic silencing of the PTEN promoter"
- Dear reviewer, we realized the modifications according to your suggestion on line 41
- "positive crosstalk between the PI3K/AKT and Wnt/β-catenin pathways" could be expressed as "reciprocal activation of the PI3K/Akt and Wnt/β-catenin pathways"
- Dear reviewer, we realized the modifications according to your suggestion on line 44
- "TOB1-AS1 regulates the activation of the Wnt/β-catenin pathway" could be improved to "TOB1-AS1 modulates the activation of the Wnt/β-catenin pathway"
- Dear reviewer, we realized the modifications according to your suggestion on line 66
- Strengths of the study:
- Comprehensive and mechanistic approach to understanding the role of HOTAIR in cervical cancer pathogenesis
- Incorporation of both in vitro experiments and in silico analyses to provide a multi-layered understanding
Limitations:
- Reliance on a single cell line for the majority of the experimental work
- Lack of discussion on the potential clinical relevance and translational applications
- Thank you for your thoughtful recommendations. We believe they have significantly enhanced our manuscript. We have added the following paragraph to discussion, dealing about small molecules to inhibit lncRNAs, specially HOTAIR. These substantive amendments cover lines 424 to 440.
The crucial role of HOTAIR in tumor development and maintenance allows us to hypothesize that, in the near future, small molecules designed through molecular docking could be utilized to inhibit the functions of this and other lncRNAs. Recent studies have demonstrated the translational potential of inhibiting HOTAIR through molecular docking with two inhibitors, including AC1NOD4Q. This inhibitor targets the 5' domain responsible for interaction with EZH2, resulting in the inhibition of metastasis in an orthotopic breast cancer model using cell lines [44]. Notably, these findings were replicated in patient-derived xenograft models, underscoring the feasibility of this approach.
In addition, the potential role of these molecules as potential biomarkers of prognosis, treatment response and therapeutic targets is becoming increasingly important. Currently, more than 39 international clinical trials are ongoing, mainly in phases I and II. In particular, HOTAIR has been registered as a biomarker in thyroid cancer. In a study involving 89 patients, HOTAIR was found to be significantly overexpressed (ΔCt value: 5.74±1.129 versus 8.22±1.301, P<0.001), underlining its potential utility as a clinical biomarker [45].
- To the authors, I commend your dedication to unraveling the complex regulatory networks governing cervical cancer progression. The insights provided in this manuscript represent a significant advancement in our understanding of the disease and open up promising avenues for future research and therapeutic development. Keep up the excellent work, and continue pushing the boundaries of our knowledge in this important field
R. Thank you very much for your words, they are very inspiring, they moved us to continue advancing in the research of lncRNAs in cancer.
Reviewer 3 Report
Comments and Suggestions for Authors
The article titled "HOTAIR Drives the Overactivation of PI3K/Akt and Wnt/β-Catenin Signaling Pathways via PTEN Hypermethylation in Cervical Cancer" makes a significant contribution to the understanding of cervical cancer (CC).
Recent research has extensively explored the role of long non-coding RNAs (lncRNAs) in cancer development, revealing their importance as regulators of key signaling pathways. LncRNAs are RNA molecules longer than 200 base pairs, transcribed primarily by RNA polymerase II, with a 5' cap and a 3' polyadenylated tail. Their complex structure enables them to interact with DNA, RNA, and proteins, allowing them to regulate transcriptional and epigenetic mechanisms. This regulatory capacity enables lncRNAs to sustain signaling pathways like PI3K/AKT and Wnt/β-catenin, thereby promoting cancer progression, including in cervical cancer.
The HOX transcription antisense intergenic RNA (HOTAIR) has been identified as a positive regulator of the PI3K/AKT and Wnt/β-catenin signaling pathways in various cancers, even in the presence of drugs and specific inhibitors targeting these pathways. Although HOTAIR's role in regulating these pathways is well-documented, the mechanisms connecting them remain largely unexplored.
This study demonstrates that HOTAIR facilitates the transcriptional inactivation of PTEN by promoting hypermethylation of its promoter region. HOTAIR achieves this by recruiting the enzyme DNMT1, thereby regulating both the PI3K/AKT and Wnt/β-catenin pathways. The authors observed decreased PTEN mRNA and protein levels in CC samples, with increased PTEN expression following HOTAIR inhibition in HeLa cells. These findings align with previous reports showing PTEN loss in up to 62% of CC cases, leading to sustained PI3K/AKT pathway activation.
In summary, this study uncovers a novel mechanism by which HOTAIR mediates positive crosstalk between the Wnt/β-catenin and PI3K/AKT signaling pathways in CC. The mechanism involves a feedback loop where HIF1α induces HOTAIR overexpression, which, in turn, sustains transcriptional activity in the Wnt/β-catenin pathway and possibly enables continued PI3K/AKT pathway activation through PTEN promoter methylation. This regulatory network highlights HOTAIR's central role in coordinating these pathways and its significance in cervical cancer progression. HOTAIR's critical regulatory role suggests it could be a promising therapeutic target in cancers characterized by PTEN downregulation.
Minor suggestion: Some of the Western blot images and figures appear stretched. Please replace them with higher-quality images.
The manuscript is well-written and draws reasonable conclusions.
Author Response
The article titled "HOTAIR Drives the Overactivation of PI3K/Akt and Wnt/β-Catenin Signaling Pathways via PTEN Hypermethylation in Cervical Cancer" makes a significant contribution to the understanding of cervical cancer (CC).
Recent research has extensively explored the role of long non-coding RNAs (lncRNAs) in cancer development, revealing their importance as regulators of key signaling pathways. LncRNAs are RNA molecules longer than 200 base pairs, transcribed primarily by RNA polymerase II, with a 5' cap and a 3' polyadenylated tail. Their complex structure enables them to interact with DNA, RNA, and proteins, allowing them to regulate transcriptional and epigenetic mechanisms. This regulatory capacity enables lncRNAs to sustain signaling pathways like PI3K/AKT and Wnt/β-catenin, thereby promoting cancer progression, including in cervical cancer.
The HOX transcription antisense intergenic RNA (HOTAIR) has been identified as a positive regulator of the PI3K/AKT and Wnt/β-catenin signaling pathways in various cancers, even in the presence of drugs and specific inhibitors targeting these pathways. Although HOTAIR's role in regulating these pathways is well-documented, the mechanisms connecting them remain largely unexplored.
This study demonstrates that HOTAIR facilitates the transcriptional inactivation of PTEN by promoting hypermethylation of its promoter region. HOTAIR achieves this by recruiting the enzyme DNMT1, thereby regulating both the PI3K/AKT and Wnt/β-catenin pathways. The authors observed decreased PTEN mRNA and protein levels in CC samples, with increased PTEN expression following HOTAIR inhibition in HeLa cells. These findings align with previous reports showing PTEN loss in up to 62% of CC cases, leading to sustained PI3K/AKT pathway activation.
In summary, this study uncovers a novel mechanism by which HOTAIR mediates positive crosstalk between the Wnt/β-catenin and PI3K/AKT signaling pathways in CC. The mechanism involves a feedback loop where HIF1α induces HOTAIR overexpression, which, in turn, sustains transcriptional activity in the Wnt/β-catenin pathway and possibly enables continued PI3K/AKT pathway activation through PTEN promoter methylation. This regulatory network highlights HOTAIR's central role in coordinating these pathways and its significance in cervical cancer progression. HOTAIR's critical regulatory role suggests it could be a promising therapeutic target in cancers characterized by PTEN downregulation.
Minor suggestion: Some of the Western blot images and figures appear stretched. Please replace them with higher-quality images.
- Dear reviewer, the corresponding modification was made on the figures.
The manuscript is well-written and draws reasonable conclusions.
Round 2
Reviewer 2 Report
Comments and Suggestions for Authors
As an experienced medical researcher with years of expertise, I offer the following feedback on the manuscript "HOTAIR promotes the hyperactivation of PI3K/Akt and Wnt/β-catenin signaling pathways via PTEN hypermethylation in cervical cancer":
The primary objective of this study is to explore the mechanisms by which the long non-coding RNA HOTAIR orchestrates the reciprocal activation of the PI3K/AKT and Wnt/β-catenin signaling pathways in cervical cancer. The authors utilized a well-designed experimental approach, including HOTAIR knockdown and overexpression, pathway inhibitors, luciferase reporter assays, and various molecular techniques such as Western blot, FISH, ChIP, and RIP. This comprehensive methodology allowed them to elucidate the regulatory axis involving HOTAIR, HIF1α, and the epigenetic silencing of PTEN.
One potential limitation of this study is the use of only two cervical cancer cell lines (HeLa and SiHa), which may limit the generalizability of the findings. Inclusion of additional cell lines or in vivo models could strengthen the conclusions.
To further enhance the quality of this manuscript, I would suggest the following:
1. Provide a more detailed discussion on the clinical implications of the HOTAIR/HIF1α/PTEN axis, and how it might inform the development of targeted therapies for cervical cancer.
2. Explore the potential crosstalk between the PI3K/AKT and Wnt/β-catenin pathways beyond the regulation by HOTAIR, as the interplay between these signaling cascades is a complex and important area of cancer biology.
3. Conduct experiments to validate the clinical relevance of the proposed regulatory mechanism, such as analyzing HOTAIR, HIF1α, and PTEN levels in a cohort of cervical cancer patient samples.
Regarding the use of English in this manuscript, a few areas could be improved:
1. "to address this knowledge gap" could be rephrased as "to address this knowledge gap in the literature".
2. "enabling them to sustain signaling pathways" could be changed to "enabling them to sustain the activation of signaling pathways".
3. "HOTAIR is central in sustaining the activation of both signaling pathways" could be written as "HOTAIR is a central regulator in sustaining the activation of both signaling pathways".
4. "our findings reveal that HOTAIR is central in sustaining the activation of both signaling pathways" could be improved to "our findings reveal that HOTAIR is a key regulator in sustaining the activation of both signaling pathways".
5. "this process leads to the activation of both pathways" could be revised to "this process leads to the sustained activation of both pathways".
The strengths of this manuscript include the comprehensive experimental approach, the novel insights into the HOTAIR/HIF1α/PTEN axis, and the potential clinical relevance of these findings. The primary limitation is the use of a limited number of cell lines, which could be addressed in future studies.
In conclusion, I commend the authors for their valuable contributions to understanding the role of HOTAIR in the dysregulation of critical signaling pathways in cervical cancer. I encourage them to continue their research in this promising area, as it may lead to the development of novel targeted therapies for this devastating disease.
Comments on the Quality of English LanguageAs an experienced medical researcher with years of expertise, I offer the following feedback on the manuscript "HOTAIR promotes the hyperactivation of PI3K/Akt and Wnt/β-catenin signaling pathways via PTEN hypermethylation in cervical cancer":
The primary objective of this study is to explore the mechanisms by which the long non-coding RNA HOTAIR orchestrates the reciprocal activation of the PI3K/AKT and Wnt/β-catenin signaling pathways in cervical cancer. The authors utilized a well-designed experimental approach, including HOTAIR knockdown and overexpression, pathway inhibitors, luciferase reporter assays, and various molecular techniques such as Western blot, FISH, ChIP, and RIP. This comprehensive methodology allowed them to elucidate the regulatory axis involving HOTAIR, HIF1α, and the epigenetic silencing of PTEN.
One potential limitation of this study is the use of only two cervical cancer cell lines (HeLa and SiHa), which may limit the generalizability of the findings. Inclusion of additional cell lines or in vivo models could strengthen the conclusions.
To further enhance the quality of this manuscript, I would suggest the following:
1. Provide a more detailed discussion on the clinical implications of the HOTAIR/HIF1α/PTEN axis, and how it might inform the development of targeted therapies for cervical cancer.
2. Explore the potential crosstalk between the PI3K/AKT and Wnt/β-catenin pathways beyond the regulation by HOTAIR, as the interplay between these signaling cascades is a complex and important area of cancer biology.
3. Conduct experiments to validate the clinical relevance of the proposed regulatory mechanism, such as analyzing HOTAIR, HIF1α, and PTEN levels in a cohort of cervical cancer patient samples.
Regarding the use of English in this manuscript, a few areas could be improved:
1. "to address this knowledge gap" could be rephrased as "to address this knowledge gap in the literature".
2. "enabling them to sustain signaling pathways" could be changed to "enabling them to sustain the activation of signaling pathways".
3. "HOTAIR is central in sustaining the activation of both signaling pathways" could be written as "HOTAIR is a central regulator in sustaining the activation of both signaling pathways".
4. "our findings reveal that HOTAIR is central in sustaining the activation of both signaling pathways" could be improved to "our findings reveal that HOTAIR is a key regulator in sustaining the activation of both signaling pathways".
5. "this process leads to the activation of both pathways" could be revised to "this process leads to the sustained activation of both pathways".
The strengths of this manuscript include the comprehensive experimental approach, the novel insights into the HOTAIR/HIF1α/PTEN axis, and the potential clinical relevance of these findings. The primary limitation is the use of a limited number of cell lines, which could be addressed in future studies.
In conclusion, I commend the authors for their valuable contributions to understanding the role of HOTAIR in the dysregulation of critical signaling pathways in cervical cancer. I encourage them to continue their research in this promising area, as it may lead to the development of novel targeted therapies for this devastating disease.
Author Response
- Provide a more detailed discussion on the clinical implications of the HOTAIR/HIF1α/PTEN axis, and how it might inform the development of targeted therapies for cervical cancer.
Answer: Dear reviewer, thank you so much for your comment. We expand the discussion on this section of the manuscript (lines 431- 450).
- Explore the potential crosstalk between the PI3K/AKT and Wnt/β-catenin pathways beyond the regulation by HOTAIR, as the interplay between these signaling cascades is a complex and important area of cancer biology.
Answer: Thank you for the observation, we argue the interplay of these pathways in lines 373-383 of the discussion section.
- Conduct experiments to validate the clinical relevance of the proposed regulatory mechanism, such as analyzing HOTAIR, HIF1α, and PTEN levels in a cohort of cervical cancer patient samples.
Answer: Thank you for your constructive feedback. We acknowledge the importance of analyzing the clinical relevance of HOTAIR regulation on the PI3K/AKT and WNT/β-Catenin pathways, and we agree that this is a significant area of investigation. However, our current work is primarily focused on elucidating the molecular mechanisms by which HOTAIR regulates gene expression in cervical cancer cells, using preclinical models. This study aims to provide foundational insights that will support future clinical applications.
We are also in the process of conducting a clinical study to examine the levels of a complex regulatory network, in which HOTAIR plays a central role. However, the scope and findings of that study warrant a separate publication. We appreciate your understanding of our current research focus and look forward to sharing the clinical implications of HOTAIR in subsequent studies.
Regarding the use of English in this manuscript, a few areas could be improved:
- "to address this knowledge gap" could be rephrased as "to address this knowledge gap in the literature".
- "enabling them to sustain signaling pathways" could be changed to "enabling them to sustain the activation of signaling pathways".
- "HOTAIR is central in sustaining the activation of both signaling pathways" could be written as "HOTAIR is a central regulator in sustaining the activation of both signaling pathways".
- "our findings reveal that HOTAIR is central in sustaining the activation of both signaling pathways" could be improved to "our findings reveal that HOTAIR is a key regulator in sustaining the activation of both signaling pathways".
- "this process leads to the activation of both pathways" could be revised to "this process leads to the sustained activation of both pathways".
Answer: Dear reviewer we welcome your editing suggestions for our writing. All of them have been attended to in the Manuscript.